# Crystal Structure of African Swine Fever Virus A179L with the Autophagy Regulator Beclin

**DOI:** 10.3390/v11090789

**Published:** 2019-08-27

**Authors:** Suresh Banjara, Gareth L. Shimmon, Linda K. Dixon, Christopher L. Netherton, Mark G. Hinds, Marc Kvansakul

**Affiliations:** 1Department of Biochemistry & Genetics, La Trobe Institute for Molecular Science, La Trobe University, Melbourne, Victoria 3086, Australia; 2Pirbright Institute, Ash Road, Pirbright, Surrey GU24 0NF, UK; 3Bio21 Molecular Science and Biotechnology Institute, The University of Melbourne, Parkville, Victoria 3050, Australia

**Keywords:** Bcl-2, Beclin, autophagy, X-ray crystallography, ASFV

## Abstract

Subversion of programmed cell death-based host defence systems is a prominent feature of infections by large DNA viruses. African swine fever virus (ASFV) is a large DNA virus and sole member of the *Asfarviridae* family that harbours the B-cell lymphoma 2 or Bcl-2 homolog A179L. A179L has been shown to bind to a range of cell death-inducing host proteins, including pro-apoptotic Bcl-2 proteins as well as the autophagy regulator Beclin. Here we report the crystal structure of A179L bound to the Beclin BH3 motif. A179L engages Beclin using the same canonical ligand-binding groove that is utilized to bind to pro-apoptotic Bcl-2 proteins. The mode of binding of Beclin to A179L mirrors that of Beclin binding to human Bcl-2 and Bcl-x_L_ as well as murine γ-herpesvirus 68. The introduction of bulky hydrophobic residues into the A179L ligand-binding groove via site-directed mutagenesis ablates binding of Beclin to A179L, leading to a loss of the ability of A179L to modulate autophagosome formation in Vero cells during starvation. Our findings provide a mechanistic understanding for the potent autophagy inhibitory activity of A179L and serve as a platform for more detailed investigations into the role of autophagy during ASFV infection.

## 1. Introduction

The inhibition of programmed cell death pathways in infected host cells is a widely used strategy employed by many large DNA viruses. Successful inhibition is often achieved by the use of virus encoded homologs of the B-cell lymphoma-2 (Bcl-2) family of proteins, which are crucial arbiters of intrinsic or mitochondrial initiated apoptosis [1]. Bcl-2 family members are distinguished into pro-survival and pro-apoptotic signalling proteins, and bear one or more hallmark Bcl-2 homology (BH) motifs that underpin their activity [2]. In higher organisms, pro-survival members of the family include Bcl-2, Bcl-x_L_, Bcl-w, Mcl-1, A1 and Bcl-b [3], although several organisms harbour unique pro-survival proteins that play organism-specific roles [4,5]. Pro-apoptotic proteins Bak and Bax are the executors of cell death in mammals by causing the release of cytochrome c from the mitochondrial outer membrane (MOM) through oligomeric pores [6,7]. The BH3-only proteins comprise the remaining pro-apoptotic Bcl-2 members and include Bid, Bim, Noxa, Puma, Bmf, Bad, Bik and Hrk. Mechanistically the BH3-only proteins act either by sequestering and neutralizing the pro-survival Bcl-2 members, or by directly activating Bak and Bax. BH3-only proteins are defined by only containing the BH3 motif, which adopts an α-helical structure to engage the canonical ligand-binding groove on the pro-survival Bcl-2 proteins [2]. In response to certain cellular insults, including exposure to cytotoxic drugs or growth factor deprivation, the BH3-only proteins are up-regulated and subsequently trigger cell death mechanisms [8].

Viruses utilize a diverse set of molecular strategies to inhibit premature host cell apoptosis. These include direct caspase inhibition and inhibition of the intrinsic apoptosis [9]. For example, herpesviruses such as Kaposi Sarcoma Herpesvirus (KSHV) or Epstein-Barr virus encode viral Bcl-2-like proteins [3,10,11,12] that are essential for successful viral replication [13]. Poxviruses encode anti-apoptotic proteins that often lack any overt sequence identity with Bcl-2. These include F1L from variola virus [14] and vaccinia virus [15,16,17,18,19,20], N1L from vaccinia virus [21,22,23], M11L from myxoma virus [24,25,26] as well as fowlpoxvirus FPV039 [27,28], canarypoxvirus CNP058 [29], orf virus ORF125 [30,31], deerpoxvirus DPV022 [32,33] and sheeppoxvirus SPPV14 [34]. Amongst the *Iridoviridae*, grouper iridovirus encodes pro-survival GIV66 [35,36].

African swine fever virus (ASFV) is the sole member of the family *Asfarviridae*, composed of large double stranded DNA viruses [37]. ASFV is the causative agent of the highly transmissible lethal haemorrhagic African swine fever infection in domestic pigs, and is endemic in East African wild pig populations [38]. ASFV was introduced into Georgia in 2007 and has since spread across Europe and Asia and is a severe threat to the global pig industry and food security, since there is no vaccine. ASFV is a complex virus, encoding for at least 150 proteins, which include an arsenal of immune modulatory proteins and virulence factors [39,40] that includes the Bcl-2 homolog A179L [41].

In cellular assays, A179L is a potent inhibitor of apoptosis, is essential to protecting both Hela cells as well as insect cells against apoptosis [42] and is localized at the mitochondria and endoplasmic reticulum [43]. A179L is an unusually promiscuous Bcl-2 protein and is able to bind to all major pro-apoptotic Bcl-2 proteins using the family-defining ligand-binding groove [44]. A179L also inhibits autophagy by binding Beclin, and prevents autophagosome formation during nutrient deprivation [43]. However, the structural basis for Beclin engagement by A179L has not been clarified. Here we report the crystal structure of A179L in complex with the BH3 domain of Beclin. Mutations in the A179L binding groove abolish Beclin binding and ablate its ability to interfere with autophagy. These findings establish a mechanistic basis for ASFV-mediated inhibition of autophagy.

## 2. Materials and Methods

### 2.1. Protein Expression and Purification

Synthetic codon-optimized cDNA encoding A179L (Uniprot Accession number P42485) was cloned into the bacterial expression vector pMAL c4x-1-M(RBS) using SacI at the 5’ end and EcoRI at the 3’ (Genscript). Recombinant A179LΔC31, a construct with the 31 C-terminal residues truncated, was expressed in BL21 DE3 Codon Plus RIPL cells using the auto-induction method [45] for 24 h at 30 °C with shaking. Bacterial cells were collected using an ultracentrifuge at 6000 rev min^−1^ (JLA 9.1000 rotor, Beckman Coulter Avanti J-E) for 20 min and resultant cell pellets were resuspended in 50 mL lysis buffer A (50 mM Tris pH 8.5, 300 mM NaCl and 5 mM BME (β-Mercaptoethanol)). The cells were lysed with sonication (Model 705 Sonic Dismembrator, Fisher Scientific, Hampton, New Hampshire, US). The lysate was transferred into SS34 tubes for further centrifugation at 16,000 rev min^−1^ (JA-25.50 rotor, Beckman Coulter Avanti J-E) for 20 min. The supernatant was loaded onto a 5 mL His Trap HP, 5 mL column (GE Healthcare, Chicago, IL, USA) equilibrated with buffer A. After sample application, the column was washed using 100 ml of buffer B (50 mM Tris pH 8.5, 300 mM NaCl, 5 mM BME (β-Mercaptoethanol) and 25 mM of imidazole). Bound protein was then eluted with buffer C (50 mM Tris pH 8.5, 300 mM NaCl, 5 mM BME (β-Mercaptoethanol) and 300 mM of imidazole) and dialysed overnight into buffer A at 4 °C. The target protein was then concentrated using a centrifugal concentrator with 30 kDa molecular weight cut-off (Amicon® Ultra 15) to a final volume of 1 mL. Concentrated A179L was subjected to size-exclusion chromatography using a Superdex S200 10/300 column mounted on an ÄKTApure system (GE Healthcare) equilibrated in 25 mM Tris pH 7.5, 150 mM NaCl with 5 mM DTT (Dithiothreital), where it eluted as a single peak. Using SDS–PAGE analysis, the final sample was determined to be 95% pure. Eluted protein was concentrated using a centrifugal concentrator with a 30 kDa molecular weight cut-off (Amicon ® Ultra 15) to a final concentration of 21 mg/mL.

### 2.2. Expression and Purification of A179L Mutants

Synthetic cDNA encoding for a codon-optimized double (V73Y/G89Y) mutant of A179L was cloned into the pGEX-6P-3 vector (Genscript). Expression and purification of wildtype A179L as well as A179L V73Y/G89Y was performed as previously described [44].

### 2.3. Measurement of Dissociation Constants

Binding affinities were measured using a MicroCal iTC200 system (GE Healthcare) at 25 °C as previously described [36]. The BH3-motif peptides used were commercially synthesized and purified to a final purity of 95% (GenScript). The sequence of the *Sus scrofa* Beclin peptide used was: DGGTMENLSRRLKVTGDLFDIMSGQT (Uniprot accession code Q4A1L5; residues 103–128). All other peptides employed were described previously [44].

### 2.4. Crystallization and Data Collection

A complex of A179L with Beclin BH3 peptide was prepared as previously described [46]. MBP-A179L was incubated with Beclin BH3 motif in a molar ratio of 1:1.2 (protein:peptide). The mixture was left on ice for 10 min followed by the addition of 4 mM of maltose and further incubating for 10 more min. High-throughput sparse matrix screening was carried out using 96-well sitting-drop trays (Swissci) and the vapour-diffusion method at 20 °C at La Trobe University, Melbourne, Australia. The initial crystallization condition used was identified using the Shotgun Screen (Molecular Dimensions). Crystals of A179L:Beclin BH3 were obtained at 20 mg ml^−1^ using the sitting-drop method at 20 °C in 0.2 M ammonium sulfate, 0.1 M Bis-Tris pH 5.5 and 25% *w/v* PEG 3350. The crystals were flash-cooled at −173 °C using 30% (*w*/*v*) glucose as cryo-protectant. Native diffraction data were collected at the Australian Synchrotron MX2 beamline using an EIGER 16M detector at a wavelength of 0.9537 Å and an oscillation range of 0.1° per frame. Data integration and scaling was performed using XDS [47] and AIMLESS [48]. The structure was solved with A179L:Bax BH3 (PDB ID 5UA5) as a search model with PHASER [49]. The structure was rebuilt manually using Coot [50] and refined using PHENIX [51]. A179L:Beclin BH3 crystals contained one chain of MBP-A179L and one chain of Beclin BH3 in the asymmetric unit, with a calculated solvent content of 51.5%, and the final model was refined to an R_work_/R_free_ of 21.4/25.5 with 96.4% of residues in the favoured region of the Ramachandran plot and no outliers. Details of the data-collection and refinement statistics are summarized in Table 1. PDB coordinates have been deposited under the accession code 6TZC at the Protein Data Bank. PyMOL Molecular Graphics System, Version 1.8 Schrödinger, LLC was used for molecular images. All software were accessed using the SBGrid suite [52]. Raw images are stored with the SBGrid Data Bank [53].

### 2.5. Autophagy Assays

Recombinant, replication deficient human adenovirus 5 (rAd) encoding full-length wildtype A179L and A179L V73Y/G89Y tagged with an N-terminal HA tag were generated using standard techniques. Vero cells were transduced with rAd, incubated for 21 h and then incubated for another 3 h in complete media or in Earles balanced salt solution to induce starvation. Cells were then fixed with methanol and stained with anti-HA (clone 3F10, Roche) and anti-LC3B (L7543, Sigma). Images were captured using a Leicia SP8 confocal microscopy and the number of LC3 puncta per cell determined for 30 cells per condition using Imaris 9.2.1. Statistical analysis was performed in MiniTab (version 18) using analysis of variance (ANOVA) plus Tukey multiple comparison test to determine statistical differences between groups.

## 3. Results

To understand the structural basis for A179L inhibition of autophagy, we determined crystal structures of A179L bound to a peptide spanning the Beclin BH3 motif (Figure 1 and Figure 2A, Table 1). Previous attempts to crystallize an A179L:Beclin BH3 complex did not yield crystals that diffracted to sufficiently high resolution, consequently we employed a maltose-binding protein fusion (MBP) of A179L to enhance crystal contact formation. Clear and continuous density was observed for MBP residues 3–367 and A179L 3–146, with the linker residues AQTNSSS presumed disordered. As shown previously, A179L adopts a Bcl-2 fold featuring 8 α-helices arranged in a globular helical bundle fold. The canonical ligand-binding groove found in other pro-survival members of the family is formed by α-Helices 2-5 and engages the BH3 motif of pro-apoptotic proteins [3]. However, the region that is the equivalent of α3 is not helical. Instead it is found in an extended configuration, mimicking the corresponding region in Bcl-x_L_ [54].

The Beclin BH3 peptide binds into a surface groove formed by helices α2-5 of A179L (Figure 2B). A superimposition of the A179L:Bid BH3 complex with the A179L:Beclin complexes results in an rmsd of 0.5 Å over 143 Cα carbon atoms (Figure 3A), indicating that the mode of BH3 motif engagement is highly similar. Beclin utilizes three canonical hydrophobic residues L110, L114 and F121 as well as T117 to engage the A179L ligand-binding groove (Figure 4) In addition to the engagement of the four conserved hydrophobic pockets in the A179L binding groove, the conserved ionic interaction between pro-apoptotic BH3 motifs and pro-survival Bcl-2 proteins formed by A179L R86 and Beclin D119 is also present. This additional ionic interaction is supplemented by further ionic interactions between A179L D80 and E76 with Beclin K115, as well as hydrogen bonds between A179L N83 and Beclin D119, A179L G85 and Beclin D122, and A179L Y46 with the main chain of Beclin L114 (Figure 4).

To examine the effect of mutations in the A179L binding groove on autophagy regulation, we generated an A179L V73Y/G89Y double mutant. Circular dichroism spectroscopy indicates that the A179L V73Y/G89Y double mutant maintains a strongly alpha helical fold identical to wildtype A179L (Appendix A). Isothermal titration calorimetry revealed that A179L V73Y/G89Y lost the ability to bind to Beclin (Figure 5). We then examined the ability of this mutant to modulate a stress-induced autophagosome formation in Vero cells. Whilst Vero cells transfected with wildtype A179L displayed only a small increase in the number of autophagosome-associated puncta, cells transfected with A179L V73Y/G89Y displayed a substantial increase in the number of puncta after starvation comparable to mock transfected cells (Figure 6, Appendix A).

## 4. Discussion

Structural and functional homologs of Bcl-2 are used by numerous large DNA viruses to subvert programmed cell death-based host defence systems. The majority of these Bcl-2 family members subvert apoptosis [1]; however, several members have also been shown to target autophagy signalling. ASFV encoded A179L has the capacity to inhibit apoptosis signalling by binding to the porcine pro-apoptotic Bcl-2 proteins Bax and Bak, Bim, Bid, Bad, Bik, Bmf, Hrk, Noxa and Puma [44,56]. In addition, A179L is also able to bind both full length Beclin [43] as well as its BH3 motif [44] and confocal microscopy experiments showed it co-localized with Beclin [43], and thus A179L harbours dual functionality to interfere with both host apoptosis and autophagy signalling.

The ability to bind the BH3 motif of Beclin has previously been reported for a number of endogenous cellular Bcl-2 proteins, including Bcl-2 [57] and Bcl-x_L_ [55], as well as for several herpesviruses encoded proteins, including Ks-Bcl-2 [58] from KSHV and M11 from murine γ68 herpesvirus [59,60] as well as adenovirus E1B19K [61]. For three of these interactions, the structural basis for Beclin engagement has been determined. A comparison of the Bcl-2:Beclin BH3 (PDB ID:5VAU), Bcl-x_L_:Beclin BH3 [55] and M11:Beclin BH3 [60] complexes with our reported A179L:Beclin BH3 complex reveals that, whilst the overall mode of binding as well as BH3 ligand engagement of the four hydrophobic pockets in the ligand binding grove and the hallmark Arg-Asp salt bridge are conserved, several differences exist at the level of individual interactions that stabilize the complexes (Figure 7).

Interestingly, all major side chain mediated ionic interactions and hydrogen bonds observed in the A179L:Beclin complex are recapitulated in Bcl-2, Bcl-x_L_ and M11 complexes with Beclin, including the hallmark ionic interaction of BH3 motifs with Bcl-2, which is found between the Beclin BH3 D119 and an Arg from Bcl-2, which is further supported by an additional hydrogen bond from Beclin D119 to a conserved Asn in Bcl-2 (Figure 3 and Figure 4). An additional ionic interaction from Beclin K117 with a Glu and Asp on Bcl-2 is also shared amongst the A179L, Bcl-2 and Bcl-x_L_ complexes, whereas in the M11:Beclin complex the recipient acidic residue is a Ser that forms a hydrogen bond with K117 instead. The unique A179L interaction with Beclin is a hydrogen bond between A179L Y46 with the main chain of Beclin L114, with the equivalent residue to Y46 in Bcl-2, Bcl-x_L_ and M11 a Phe. The high level of similarity between the A179L, Bcl-2 and Bcl-xL complexes with Beclin is also reflected in the affinities of these interactions, which are comparable with A179L binding Beclin with a K_D_ of 1.9 μM, with the corresponding K_D_ for Bcl-2 being 1.7–8.0 μM [56,57] and for Bcl-x_L_ being 1.1–2.3 μM [7,51,56]. Conflicting data exists for the M11:Beclin BH3 interaction, with both a considerably tighter K_D_ of 40 nM and a weaker one of 1.1 μM being reported [56,57]. However, despite the potentially substantial difference in affinity between A179L and M11 for Beclin, both viral proteins are able to interfere with autophagosome formation.

Autophagy is an intracellular bulk degradation pathway that is conserved in eukaryotic cells and in the last decade has been implicated in a range of cellular processes. Autophagy plays an important role in the innate and adaptive immune response to infection and can directly degrade invading pathogens [62,63]. However, certain viruses hijack the autophagy pathway to benefit replication [64]. Little is known about the role of autophagy during ASFV infection beyond the observation that A179L can inhibit the formation of starvation-induced autophagosomes, although the major autophagosome structural protein LC3B localized to ASFV replication sites when overexpressed [43]. Future experiments should focus on the functional significance of the role of autophagy, and the modulation of the pathway by A179L, in ASFV replication. Of course it is important to note that ASFV replicates in both mammalian and arthropod hosts and membranes that resemble autophagosomes have been observed enveloping ASFV virions in ticks [65]. Modulation of autophagy by A179L may contribute to the persistence of ASFV in the arthropod host [66].

Introduction of the large hydrophobic residue Tyr in two locations within the binding groove of A179L ablates the ability of A179L to bind to Beclin and its ability to inhibit the formation of autophagosomes during starvation. A previously reported mutant of A179L, A179L G89A, was shown to no longer bind promiscuously to BH3 motifs from all major pro-apoptotic Bcl-2 proteins and Beclin, and instead only bound Puma BH3 [44], indicating that it is indeed possible to engineer single-ligand specificity into the A179L binding groove. Consequently, it may be possible to engineer mutants of A179L that discriminate between its autophagy and apoptosis inhibitory function. K115 in Beclin is engaged in two ionic interactions with A179L D80 and E76, and mutation of both residues in A179L is likely to substantially reduce Beclin binding. Conversely, mutation of A179L L50, a residue located at the base of the binding pocket that accepts Beclin Thr 117 for a larger hydrophobic residue, would be predicted to favour Beclin binding over binding to pro-apoptotic Bcl-2 members that have larger residues at the equivalent position to Beclin Thr 117.

In summary, we report the structural basis for A179L-mediated inhibition of autophagy by determining the crystal structure of A179L bound to the Beclin BH3 motif. Furthermore, we show that disruption of Beclin binding via targeted mutations in the A179L ligand-binding grove ablates the ability of A179L to suppress autophagosome formation. These findings provide a mechanistic platform for more detailed investigations into the role of A179L during autophagy inhibition and to delineate the relative contributions that A179L-mediated suppression of apoptosis and autophagy signaling makes during ASFV infection and viral persistence.

## Figures and Tables

**Figure 1 viruses-11-00789-f001:**
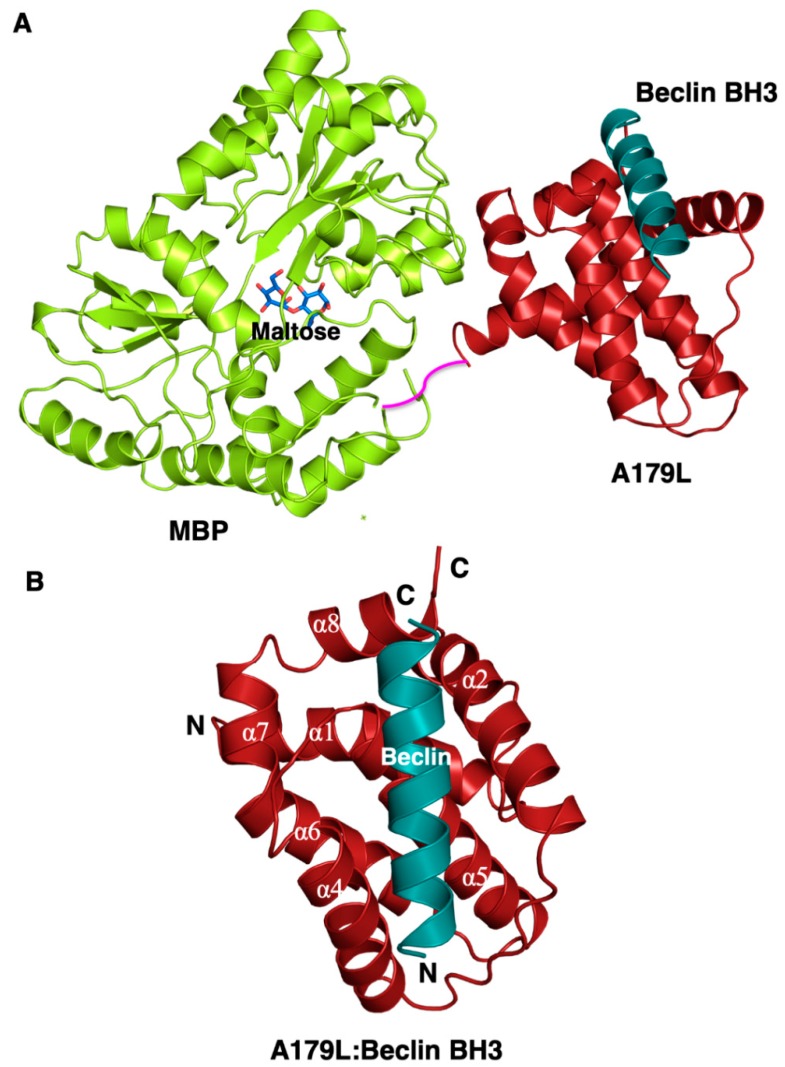
Crystal structure of A179L:Beclin BH3. (**A**) Crystal structure of maltose-binding protein (MBP) fusion (green limon) in complex with maltose (blue sticks) fused at the N-terminus of A179L (red firebrick) in complex with a peptide spanning the BH3 motif of Beclin (cyan). A short linker comprising the residues NSSS lacking electron density between MBP and A179L is shown in magenta and was modelled by hand. (**B**) The conserved Bcl-2 fold of A179L comprising 8 α-helices with the Beclin BH3 peptide bound in the hydrophobic groove formed by α2-5. Helix names have been retained to be identical to those of the Bcl-x_L_:BH3 Beclin complex (PDB ID:2P1L) [55].

**Figure 2 viruses-11-00789-f002:**
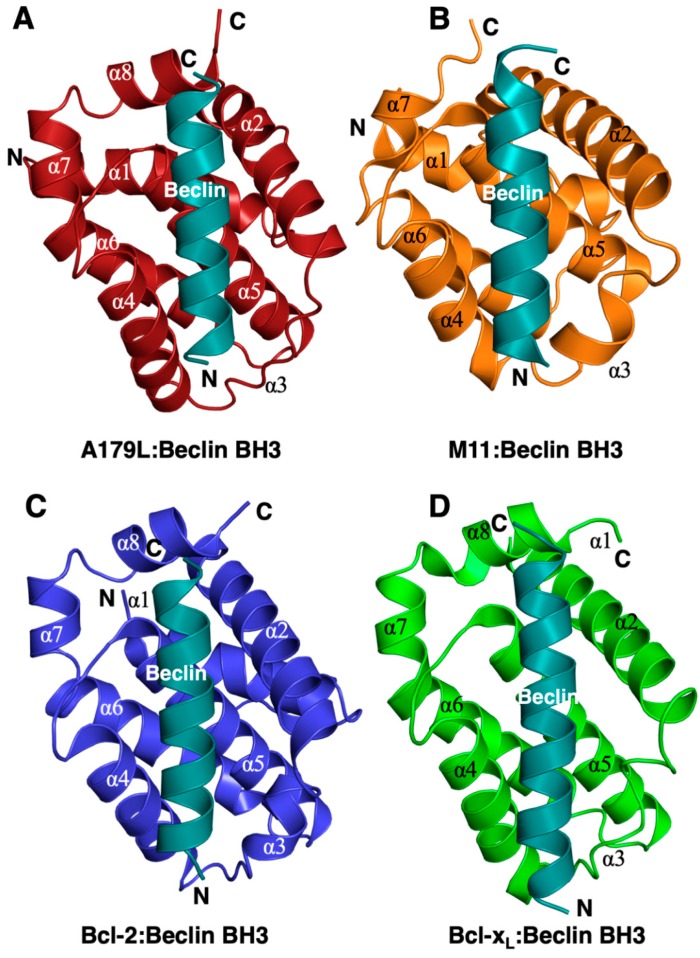
Comparison of the structure of A179L:Beclin BH3 with other Beclin BH3 complexes of Bcl-2 proteins. Ribbon representation of (**A**) A179L (firebrick) in complex with the swine Beclin BH3 motif (cyan). A179L helices are labeled α1-8. The view in (A) shows the hydrophobic binding groove formed by helices α2-5. (**B**) γ-herpesvirus 68 M11 (orange) in complex with the human Beclin BH3 motif (cyan) (PDB ID:3BL2). (**C**) Human Bcl-2 (blue) in complex with the human Beclin BH3 motif (cyan) (PDB ID:5VAU). (**D**) Human Bcl-x_L_ (green) in complex with human Beclin BH3 motif (cyan) (PDB ID:2P1L). All views in (**B**–**D**) are as in (**A**). The orientation of the ribbons is identical to Figure 1B and the structures of Bcl-x_L_:Beclin BH3, Bcl-2:Beclin BH3 and M11 Beclin BH3 were aligned on A179L Beclin BH3 using Coot [50].

**Figure 3 viruses-11-00789-f003:**
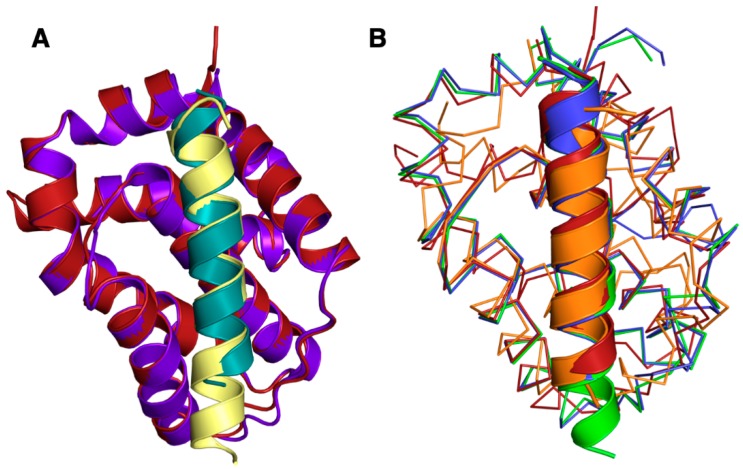
Superimposition of A179L:Beclin with A179L:Bid and other pro-survival Bcl-2:Beclin complexes. (**A**) Cartoon view of A179L (firebrick):Beclin (teal) superimposed with A179L (purple):Bid (yellow) (PDB ID 5UA4). (**B**) A179L:Beclin superimposed with M11:Beclin (orange), Bcl-X_L_:Beclin (green) and Bcl-2:Beclin (blue). Beclin is shown as cartoon, whereas A179L, Bcl-2, Bcl-x_L_ and M11 backbones are shown as Cα trace. The orientation of the traces is identical to Figure 2 and the structures of Bcl-x_L_:Beclin BH3, Bcl-2:Beclin BH3 and M11 Beclin BH3 were aligned on A179L Beclin BH3 using Coot [50].

**Figure 4 viruses-11-00789-f004:**
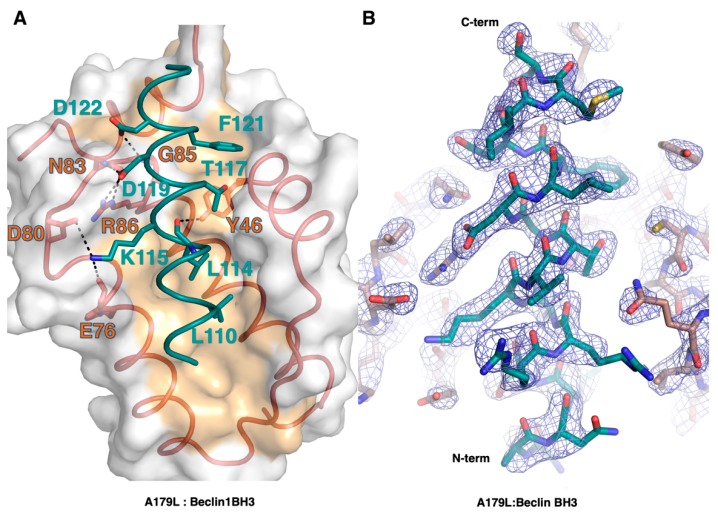
Detailed view of the A179L:BH3 peptide interfaces. (**A**) A179L is shown as a grey molecular surface, with the backbone and floor of the binding groove shown in green and orange, respectively. Beclin BH3 is shown in cyan. The four conserved hydrophobic residues of Beclin BH3 (L110, L114, T117, F121) engage the binding groove and the family-defining salt bridges formed by A179L R86 and Beclin D119 are labeled. Additional ionic interactions and hydrogen bonds are shown as black dots. (**B**) 2Fo-Fc electron density map (blue mesh) of Beclin BH3 peptide (cyan) bound to A179L (orange). Map is contoured at σ1.5.

**Figure 5 viruses-11-00789-f005:**
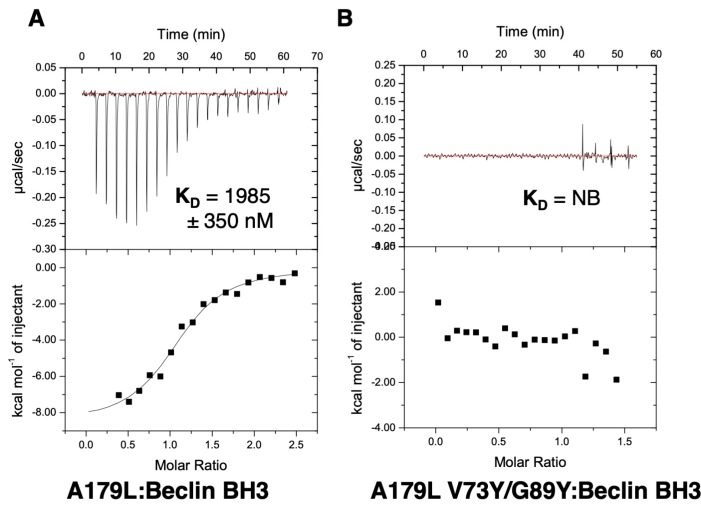
Isothermal titration calorimetry-binding profiles of (**A**) A179L as well as (**B**) A179L V73Y/G89Y interaction with Beclin BH3 peptide. Raw thermogram and a binding isotherm fitted with a one-site binding model are shown. All experiments were performed in triplicate. Data for wildtype A179L are from [44]. K_D_: Dissociation constant; ±: Standard deviation; NB: No binding.

**Figure 6 viruses-11-00789-f006:**
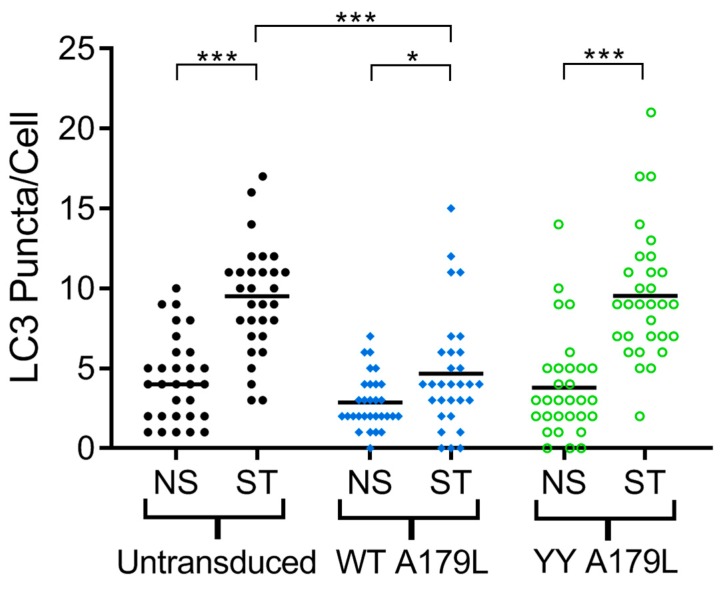
A179L-mediated inhibition of autophagosome formation. The number of LC3 puncta per cell for 30 individual cells per indicated experimental condition was quantified by Imaris analysis of confocal images. Vero cells were transduced with AdH5 vectors encoding either wildtype A179L or A179L V73Y/G89Y, or left untransduced and were incubated for a total of 24 h. Prior to fixation, cells were either incubated in complete cell media (NS) or starved in EBSS (ST) for 3 h to induce autophagy. Centre lines show the medians and asterisks represent significant differences between the values (* *p* < 0.05, *** *p* < 0.001).

**Figure 7 viruses-11-00789-f007:**
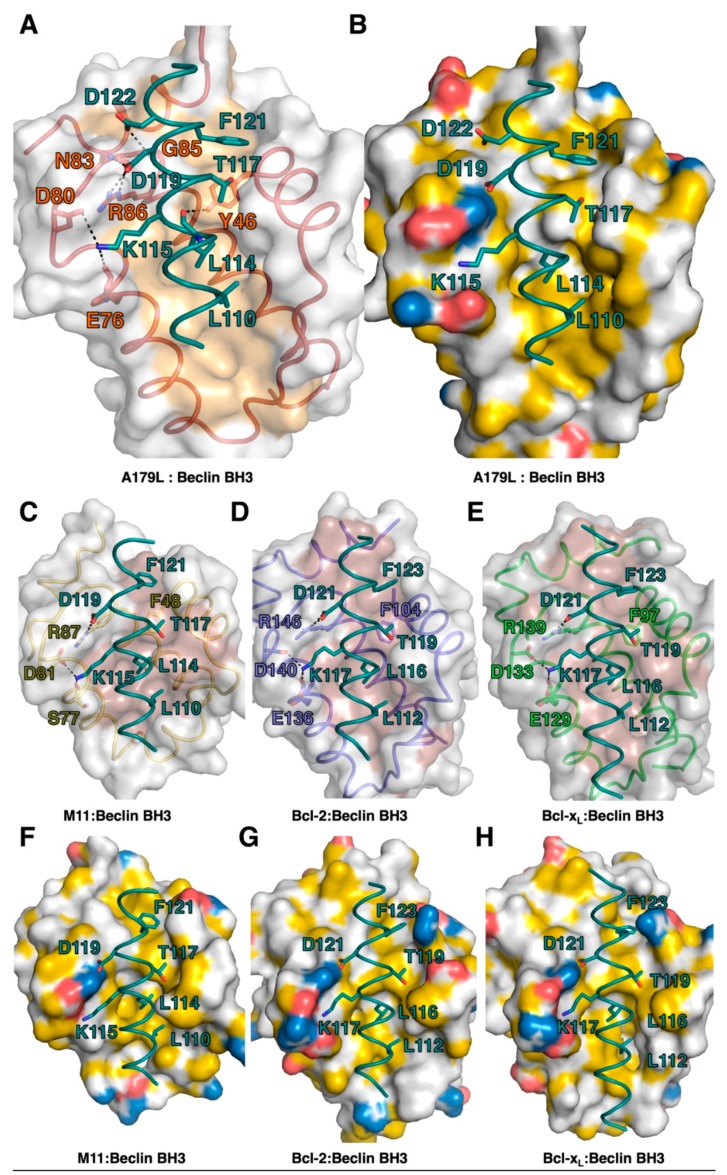
Detailed view of the interfaces of pro-survival Bcl-2 protein complexes with Beclin BH3. (**A**) A179L is shown as a grey molecular surface, with the backbone and floor of the binding groove shown in green and orange, respectively. Beclin BH3 is shown in cyan. (**B**) Surface representation of A179L bound to Beclin BH3. A179L surface is shown in grey, charged residues are shown in blue and red, hydrophobic residues in yellow. Beclin BH3 is shown as cyan sticks. (**C**) γ-herpesvirus 68 M11 (orange) in complex with the human Beclin BH3 motif (cyan) (PDB ID:3BL2). (**D**) Human Bcl-2 (blue) in complex with the human Beclin BH3 motif (cyan) (PDB ID:5VAU). (**E**) Human Bcl-x_L_ (green) in complex with human Beclin BH3 motif (cyan) (PDB ID:2P1L). All views in (C)–(E) are as in (A). Hydrogen bonds and ionic interactions are shown as black dotted lines. (**F**) Surface representation of M11 bound to Beclin BH3. M11 surface is shown in grey, charged residues are shown in blue and red, hydrophobic residues in yellow. Beclin BH3 is shown as cyan sticks. (**G**) Surface representation of Bcl-2 bound to Beclin BH3. M11 surface is shown in grey, charged residues are shown in blue and red, hydrophobic residues in yellow. Beclin BH3 is shown as cyan sticks. (**H**) Surface representation of Bcl-x_L_ bound to Beclin BH3. M11 surface is shown in grey, charged residues are shown in blue and red, hydrophobic residues in yellow. Beclin BH3 is shown as cyan sticks.

**Table 1 viruses-11-00789-t001:** Crystallographic data collection and refinement statistics.

	A179L:Beclin BH3
**Data collection**	
Space group	P 2_1_
Cell dimensions	
*a*, *b*, *c* (Å)	54.56 44.34 129.02
α, β, γ (°)	90, 94.53, 90
Resolution (Å)	51.59–2.41 (2.47–2.41) *
*R*_sym_ or *R*_merge_	0.12 (0.52)
*I/*σ*I*	4.60 (1.40)
Completeness (%)	99.40 (99.70)
Multiplicity	2.9 (2.7)
CC1/2	0.98 (0.48)
**Refinement**	
Resolution (Å)	48.74–2.40 (2.49–2.41) *
No. reflections	24020 (2373)
*R*_work_/*R*_free_	0.214/0.255
Molprobity clashscore	1.63
No. atoms	
Protein	4081
Ligand/ion	23
Water	107
*B*-factors	
Protein	42.56
Ligand/ion	44.98
Water	40.29
R.m.s. deviations	
Bond lengths (Å)	0.013
Bond angles (°)	1.5

* Values in parentheses are for highest-resolution shell.

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
