# Peer review of "Crystal Structure of African Swine Fever Virus A179L with the Autophagy Regulator Beclin"

_viruses, 2019, doi:10.3390/v11090789_

Round 1

Reviewer 1 Report

The paper “Crystal Structure of African Swine Fever Virus A179L with the Autophagy Regulator Beclin” is clear and logical continuation on the research article published in 2017 in the Journal of Virology “Structural Insight into African Swine Fever Virus A179L-Mediated Inhibition of Apoptosis”. Authors solved crystal structure of BCL-2 analogue from Swine Fever Virus A179L in complex with the Beclin BH3 motif. Comparison of the motif binding mode with its binding in γ-herpesvirus 68 M11, Human Bcl-2, Bcl-xL shows many similarities but some differences, emphasized in conclusions. ITC experiment gave the binding affinity comparable with those in γ-herpesvirus 68 M11, Human Bcl-2, Bcl-xL. In addition to above authors generated double mutant incapable of Beclin BH3 motif binding and proved that opposite to cells overexpressing A179L such mutant do not have protective effect on the starving cells.

Minor comments

Page 3 line 123. Please add PDB deposition information instead of “XXX”. Page 7. Figure 3. Part A could be extended with the details of the interfaces in γ-herpesvirus 68 M11, Human Bcl-2, Bcl-xL to facilitate the statement in the conclusions on page 9 lines 231-234 on similarities and differences for the readers unfamiliar with γ-herpesvirus 68 M11, Human Bcl-2, Bcl-xL structures. Is it any charge complementarity between the Beclin BH3 motif and the binding groves in compared proteins?

Author Response

"Page 3 line 123. Please add PDB deposition information instead of “XXX”. Page 7."

We have now included the PDB accession code, which is 6TZC.

"Figure 3. Part A could be extended with the details of the interfaces in γ-herpesvirus 68 M11, Human Bcl-2, Bcl-xL to facilitate the statement in the conclusions on page 9 lines 231-234 on similarities and differences for the readers unfamiliar with γ-herpesvirus 68 M11, Human Bcl-2, Bcl-xL structures. Is it any charge complementarity between the Beclin BH3 motif and the binding groves in compared proteins?" 

We have now included detailed views of the interfaces of M11, Bcl-2 and Bcl-xL complexes with Beclin in a new Figure 7. These new panels also show the charge complementarity between Beclin BH3 and the binding grooves of A179L whose structures have been determined when bound to Beclin BH3. Binding of the BH3 motif is largely driven by hydrophobic interactions, however several ionic interactions can be seen that contribute to the overall affinity.

Reviewer 2 Report

This manuscript describes a novel structure between the protein A179L of asfavirus and the BH3 domain of Beclin.

The overall binding-mode is expected and close resembles published structures of Bcl-2 like proteins with Beclin. However, this is the first structure of this complex for asfavirus and reveals differences with previous structures. An illustration of the use of this novel structure is provided by structure-based mutagenesis of two residues, which disrupts the formation of the complex. Similar approaches are likely to provide further tools to investigate viral inhibition of autophagy.

I have two specific comments:

what is the evidence that the double-mutant protein is folded? does the structure provide ways to inhibit the anti-autophagy function without affecting the anti-apoptotic role of A179L? It would be nice to see this point discussed in more details for the existing mutant and future ones.

Author Response

"What is the evidence that the double-mutant protein is folded?"

We have added circular dichroism spectroscopy data for both wildtype and double mutant A179L that display the characteristics of a highly alpha helical protein. Furthermore, they reveal little differences between the two proteins, suggesting that the double mutant maintains the same fold as the wild type protein.

"Does the structure provide ways to inhibit the anti-autophagy function without affecting the anti-apoptotic role of A179L? It would be nice to see this point discussed in more details for the existing mutant and future ones." 

The structure does suggest avenues to design selective mutants that may selectively inhibit either the apoptosis or the autophagy response of the infected host cell, and such an approach is likely to reveal additional facets of the interplay between apoptosis and autophagy during ASFV infection. We have added the following paragraph to the discussion on page 11: “Introduction of the large hydrophobic residue Tyr in two locations within the binding groove of A179L ablates the ability of A179L to bind to Beclin, and its ability to inhibit the formation of autophagosomes during starvation. A previously reported mutant of A179L, A179L G89A, was shown to no longer bind promiscuously to BH3 motifs from all major pro-apoptotic Bcl-2 proteins and Beclin, and instead only bound Puma BH3 [41], indicating that it is indeed possible to engineer single-ligand specificity into the A179L binding groove. Consequently it may be possible to engineer mutants of A179L that discriminate between its autophagy and apoptosis inhibitory function. K115 in Beclin is engaged in two ionic interactions with A179L D80 and E76, and mutation of both residues in A179L is likely to substantially reduce Beclin binding. Conversely, mutation of A179L L50, a residue located at the base of the binding pocket that accepts Beclin Thr 117 for a larger hydrophobic residue would be predicted to favour Beclin binding over binding to pro-apoptotic Bcl-2 members that have larger residues at the equivalent position to Beclin Thr117.”.